# An Oligocene giant rhino provides insights into *Paraceratherium* evolution

Tao Deng [1,2,3✉], Xiaokang Lu[4], Shiqi Wang[1,2], Lawrence J. Flynn[5], Danhui Sun[1,3], Wen He[6] & Shanqin Chen[6]

As one of the largest land mammals, the origin and evolution of the giant rhino *Paraceratherium bugtiense* in Pakistan have been unclear. We report a new species *Paraceratherium linxiaense* sp. nov. from northwestern China with an age of 26.5 Ma. Morphology and phylogeny reveal that *P. linxiaense* is the highly derived species of the genus *Paraceratherium*, and its clade with *P. lepidum* has a tight relationship to *P. bugtiense*. Based on the paleogeographical literature, *P. bugtiense* represents a range expansion of *Paraceratherium* from Central Asia via the Tibetan region. By the late Oligocene, *P. lepidum* and *P. linxiaense* were found in the north side of the Tibetan Plateau. The Tibetan region likely hosted some areas with low elevation, possibly under 2000 m during Oligocene, and the lineage of giant rhinos could have dispersed freely along the eastern coast of the Tethys Ocean and perhaps through some lowlands of this region.

[1] Key Laboratory of Vertebrate Evolution and Human Origins, Institute of Vertebrate Paleontology and Paleoanthropology, Chinese Academy of Sciences, Beijing, China. [2] CAS Center for Excellence in Life and Paleoenvironment, Beijing, China. [3] University of Chinese Academy of Sciences, Beijing, China. [4] Henan University of Chinese Medicine, Zhengzhou, Henan, China. [5] Department of Human Evolutionary Biology, Harvard University, Cambridge, MA, USA. [6] Hezheng Paleozoological Museum, Hezheng, Gansu, China. ✉email: dengtao@ivpp.ac.cn

The giant rhino (derived genera of Paraceratheriidae, Rhinocerotoidea, Perissodactyla, such as *Paraceratherium*, *Dzungariotherium*, *Aralotherium*, and *Turpanotherium*) has been considered as one of the largest land mammals that ever lived[1]. Its skull and legs are longer than all reported land mammals, but the metapodials are not massive in outline. Its body size was suitable for open woodlands under humid or arid climatic conditions[2]. Except for some remains found in Eastern Europe[3], Anatolia[4,5], and Caucasus[6], giant rhinos lived mainly in Asia, especially in China, Mongolia, Kazakhstan, and Pakistan[7]. All forms of the giant rhino, including six genera, have been recorded from northwest to southwest China through the middle Eocene to the late Oligocene. Among them, *Juxia* from the middle Eocene is considered as the unequivocal ancestor of all giant rhinos because of a set of the primitive and primarily specialized features[7]. The genus *Paraceratherium* was the most widely distributed form of the giant rhino, but aside from East and Central Asia, many records from East Europe[3] and West Asia[4–6] comprise fragmentary specimens, and only *Paraceratherium bugtiense*, known from the southwestern corner of the Tibetan Plateau[8], has ample records and undoubtable taxa identity and is key to the origin and dispersal history of *Paraceratherium*[9,10].

Here we report a completely preserved skull with an articulated mandible and atlas, representing a new species of *Paraceratherium* from the upper Oligocene Jiaozigou Formation of the Linxia Basin in Gansu Province, China, located at the northeastern border of the Tibetan Plateau[11,12]. The Jiaozigou fauna of the Linxia Basin also includes the giant rhinos *Turpanotherium* and *Dzungariotherium*, the rodent *Tsaganomys*, the creodont *Megalopterodon*, the chalicothere *Schizotherium*, the hyracodont *Ardynia*, the rhinocerotid *Aprotodon*, and the entelodont *Paraentelodon* (Supplementary Table 1), making it similar to the Nanpoping fauna of the Lanzhou Basin and other Tabenbulukian faunas from Inner Mongolia and Ningxia. This suggests the widespread occurrence of open woodland during the late Oligocene in northwestern China, with a mix of woodland and grassland[7].

## Results

**Systematic paleontology**. Perissodactyla Owen, 1848

 Paraceratheriidae Osborn, 1923

 *Paraceratherium* Forster-Cooper, 1911

 *Paraceratherium linxiaense* sp. nov.

**Type specimens**. A complete skull and mandible with the associated atlas (holotype, HMV 2006, Fig. 1), and an axis and two thoracic vertebrae of another individual (paratype, HMV 2007, Fig. 2), which are preserved at the Hezheng Paleozoological Museum in Hezheng County, Gansu Province, China. HMV 2006 represents a full adult individual. The specific name, linxiaense, refers to the geographical location of the discovery in the Linxia Basin (Fig. 3).

**Type locality and horizon**. IVPP locality LX1808 (N35°35′05.16", E103°18′51.02"; 1983 m above sea level, Fig. 4) is near the village of Wangjiachuan, 10.8 km southwest of the town of Dongxiang County, Linxia Hui Autonomous Prefecture, Gansu Province, China (Fig. 3). HMV 2006 and 2007 are from the sandstones in the lower part of the Jiaozigou Formation (Fig. 5, Supplementary Note 1).

**Age**. IVPP locality LX1808 is faunally and paleomagnetically dated to the middle of chron C8r with an estimated age of 26.5 million years ago (Ma) in the late Oligocene (Fig. 5).

**Diagnosis**. *Paraceratherium linxiaense* possesses features that characterize the genus, such as a giant body size, long premaxillae with anterior ends extending downward, separated parietal crests, high condyle compared to the height of nuchal surface, lower inferior border of the posttympanic process than the condyle, roughly horizontal anterior part of symphysis, and downward turning cone-shaped I1. It is more derived than other species within this genus in having a larger body size, deeper nasal notch above M2, much higher occipital part and posterior end of zygomatic arch, and smaller upper incisor I1. The lower margin of the horizontal mandibular ramus is concave under the diastema, and small i1 extends anteriorly and horizontally. The dental formula is 1.0.3.3/1.0.3.3. P2 is semimolarized, while P3 and P4 are submolarized. The metaconule connects with the ectoloph and the anterior point of the hypocone in moderate wear; the antecrochet is moderate; the lingual border of the protocone is rounded on molars; and the ecto-posterior corner of the protolophid is angular on p3 and p4. The atlas has an expanded transverse foramen and a dumb-bell shaped vertebral fossa.

**Comparative description**. The new species differs from other species of *Paraceratherium* (*P. grangeri*, *P. huangheense*, *P. asiaticum*, *P. bugtiense*, and *P. lepidum*) in having a deeper nasal notch whose bottom is located above the middle of M2, proportionally larger height of the condyle (43.9%) compared to height of the occipital surface (Supplementary Table 2), short muzzle bones and diastema anterior to cheek teeth, highly raised occiput, and high zygomatic arch with a prominent posterior end (Fig. 1).

The nasals of *P. linxiaense* are flat and straight, and the nasal notch is very deep above the middle of M2 with a short distance from the orbit (15.3% of the basal cranial length) (Supplementary Table 2), much deeper than those of other species of *Paraceratherium* except *P. lepidum*, indicating a short prehensile nose trunk. The dorsal surface of the skull is shallowly depressed, different from the domed skull of *P. grangeri*[13] or the flat one of *P. lepidum*[7]. The distance between the parietal crests is narrow and smaller than that of *P. lepidum*. The infraorbital foramen is situated above the P4/M1 boundary, and the anterior margin of the orbit is located above the M2/M3 boundary. Both characters are similar to those of *P. lepidum* and more posteriorly positioned than those of *P. grangeri* above the P3/P4 boundary and the middle of M2 respectively. The position of the zygomatic arch is high, posteriorly flush with the upper margin of the orbit like those of *P. grangeri* and *P. lepidum*, and much higher than that of *P. bugtiense*[14]. The postorbital process is absent. The space between the posttympanic and postglenoid processes is moderate as in *P. lepidum*, wider than in *P. grangeri* and narrower than in *P. bugtiense*. The posttympanic process has no transverse expansion, and its lateral margin is almost flush with the postglenoid process. The posttympanic and paraoccipital processes fuse to become a wide and thick plate.

For *P. linxiaense*, the posterior border of the mandibular symphysis is situated at the p4/m1 boundary, and the posterior margin of the ascending ramus is anteriorly inclined, different from vertical situation in *P. bugtiense* and *P. lepidum*. The vascular notch of lower margin of the mandible is deep but more anterior than that of *P. lepidum*. The mandibular diastema has a straight and slowly declining upper margin like in *P. grangeri*, while strongly declined in *P. asiaticum* and *P. lepidum*, and convex in *P. huangheense* and *P. bugtiense*. The mental foramen is situated under the p3/p4 boundary, more posteriorly than in *P. asiaticum*, *P. bugtiense*, and *P. lepidum* where is under p2.

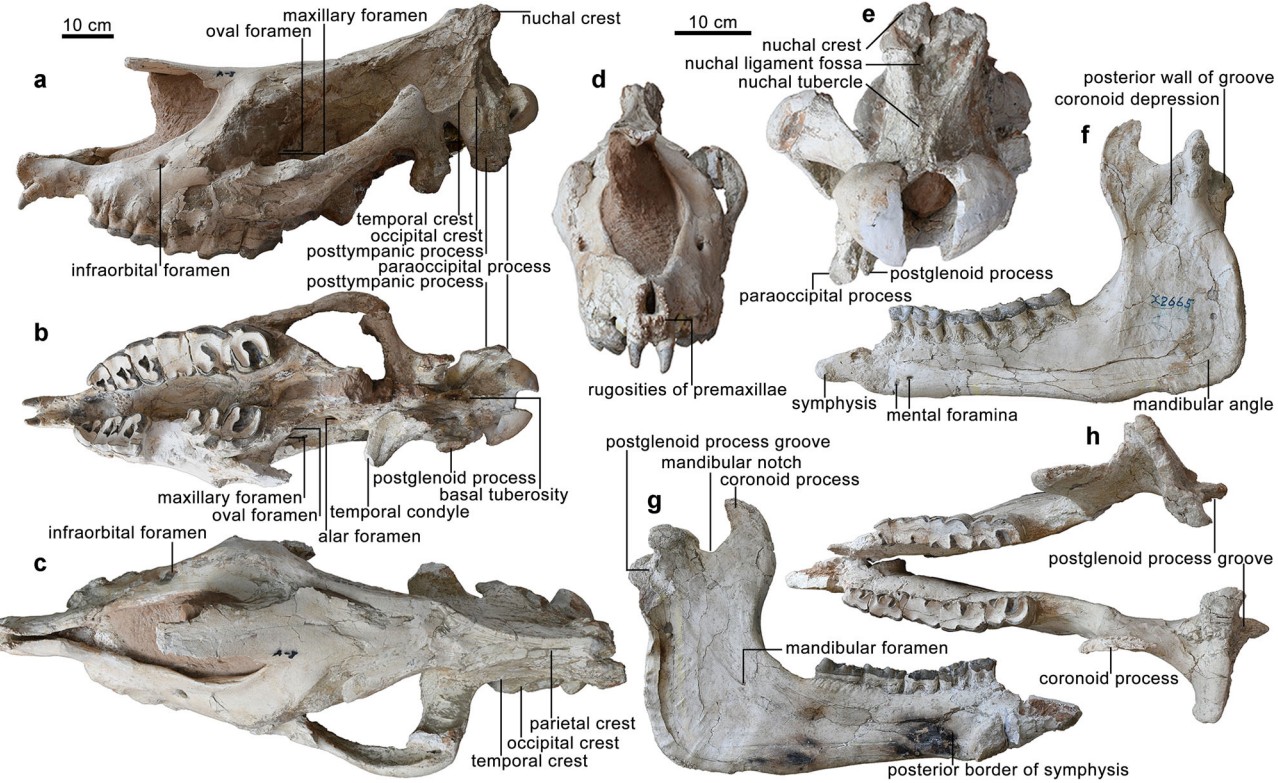

**Fig. 1 Holotype (HMV 2006) of *Paraceratherium linxiaense* sp. nov.** Skull: **a** lateral view; **b** ventral view; **c** dorsal view; **d** anterior view; **e** occipital view. Mandible: **f**, **h** lateral view and medial view of left ramus, respectively; **g** occlusal view. Skull and mandible share the scale bar, but both anterior and nuchal views have an independent scale bar.

The distance between the anterior margins of I1 and P2 is 164 mm. DP1 is absent like in most species of *Paraceratherium*, but present in *P. grangeri*. The paracone rib is absent from P2 to M1, weak in M2, and marked in M3, which is the common character of the genus *Paraceratherium*, much different from the strong paracone rib of the primitive giant rhino *Forstercooperia*[15]. The occlusal surface of P2 is triangular in *P. linxiaense*, different from the trapezium outline in *P. grangeri* and *P. asiaticum*. There is an obvious separating groove between the protocone and hypocone of P3 and P4. The hypocone of P4 is situated behind the metaconule as in *P. grangeri* and *P. lepidum*, and it is expanded and rounded, while that of *P. lepidum* is square. The antecrochet on upper molars is larger than those of *P. grangeri* and *P. asiaticum*. M3 is triangular in occlusal outline, similar to those of *P. bugtiense* and *P. lepidum*, but different from the rectangular outline in *P. grangeri* and *P. asiaticum*.

*P. linxiaense* has more reduced i1. The p2 is small and single-rooted, but large and double-rooted in *P. grangeri*. The p3 and p4 have a rudimentary entolophid, while the entoconid of *P. asiaticum* is an isolated cusp. The entolophid of m3 is nearly transverse.

Additional descriptions, measurements and comparisons, such as information of the postcranial bones, are provided in Supplementary Note 2 and Supplementary Tables 1–6.

## Discussion

With the previous studies[7,16] as a basis, the systematics of paraceratheres has been partially resolved at the genus level. These works suggest that *Juxia* diverged from the *Forstercooperia-Pappacera* clade at about 40 Ma in the middle Eocene, and its stock evolved into *Urtinotherium* in the late Eocene and the derived genus *Paraceratherium* in the Oligocene. However, the phylogenetic relationships among other derived genera, including the late

Eocene to early Oligocene *Aralotherium* and the late Oligocene *Dzungariotherium* and *Turpanotherium*, and among the species within the genus *Paraceratherium* has not been revealed through cladistics analysis.

Our phylogenetic analysis of 155 characters, including 73 newly added cranial, dental, and postcranial characters, coded from 11 giant rhinos and 16 other Rhinocerotoidea taxa (including 2 extant), yield a single most parsimonious tree with length 327, consistency index 0.60, and retention index 0.79 (Fig. 6), which places *P. linxiaense* as a derived giant rhino, nested within the monophyletic clade of the Oligocene Asian *Paraceratherium*. In the most parsimonious trees (Supplementary Figs. 1, 2), *Paraceratherium* has a relationship closer to late Eocene *Aralotherium* than to *Dzungariotherium* and *Turpanotherium*[7]. The latter two late Oligocene genera formed a clade. Like *Paraceratherium*, *Aralotherium* continues the tendency to enlarge the lower incisor i1 of the giant rhino to the maximum level, but its incisor is implanted in an anteriorly downward bent symphysis, which is derived from the straight stage seen in *Urtinotherium* and *Paraceratherium*. The most outstanding morphology of the late Eocene *Aralotherium*, as well the largest giant rhino *Dzungariotherium* lies in having a nasal notch extending posteriorly to over and above the anterior rim of the orbit. This extremely specialized nasal notch is more specific to the giant rhino. Within Rhinocerotoidea, evidently, *Paraceratherium* is at a moderate level for this specialization, which is further demonstrated by the less derived cheek teeth and conservative postcranial bones. Nevertheless, it remains uncertain whether the relatively smaller body size and incisors in *Aralotherium* and *Turpanotherium* are retained from the primitive ancestor or reduced from larger paraceratheres.

Within the *Paraceratherium* clade, our phylogenetic analysis produces a series of progressively more-derived species from

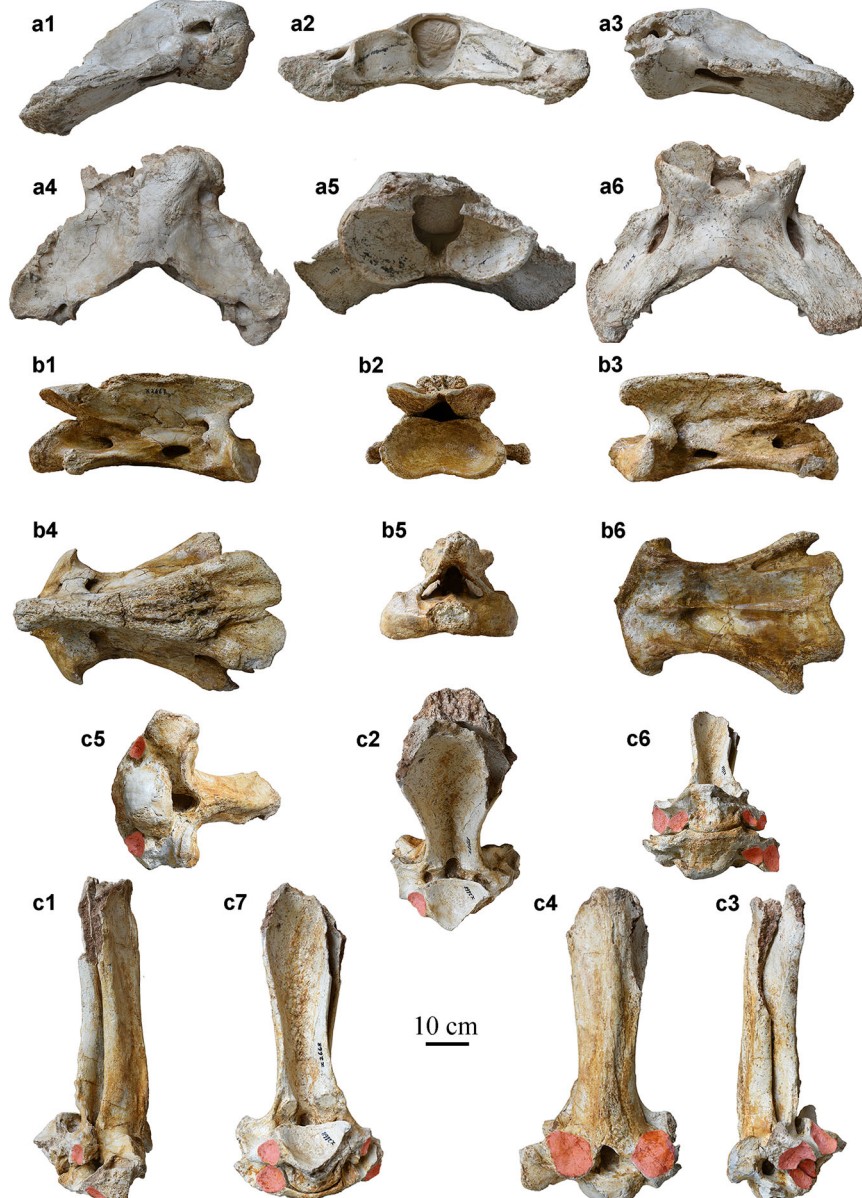

**Fig. 2 Vertebrae of *Paraceratherium linxiaense*. a** atlas (HMV 2006); **b** axis (HMV 2007); **c** 4th–5th articulated thoracic vertebrae (HMV 2007). 1, right view; 2, posterior view; 3, left view; 4, dorsal view; 5, anterior view; 6, ventral view; 7, postero-ventral view. The articular facets on 4th–5th thoracic vertebrae are marked with the red semitransparent outline whatever it is intact or has been damaged.

*P. grangeri*, through *P. huangheense*, *P. asiaticum*, and *P. bugtiense*, and terminating in *P. lepidum* and *P. linxiaense* (Fig. 6) (see Methods and Supplementary Data 1–3 for all details of the analysis). *Paraceratherium linxiaense* exhibits a high level of specialization, similar to *P. lepidum*, and both the concerned clade and *P. bugtiense* derive from a common ancestor (*P. linxiaense* has slender skull and cervical vertebrae). This hypothesis is strongly supported by the morphology of mandible. A notch on the lower margin of the mandible in front of the mandibular angle occurred firstly in *P. bugtiense*, where it is shallow and concave, and then evolved as a deeper and wider notch in *P. linxiaense* and *P. lepidum*. In addition, the latter two species have a much larger body, for example, the mandibular length of *P. bugtiense* is 4/5 of *P. lepidum*[7] or *P. linxiaense*. *P. lepidum* and *P. linxiaense* have different directions of specialization. Within *Paraceratherium*, upturning of the lower margin of the symphysis has a tendency of gradually backward reduction. It is at the level of p2 in *P. grangeri* and *P. huangheense*,

at p3 in *P. bugtiense* and *P. linxiaense*, but at p4 in *P. lepidum*[7,17–19]. Adaptation of the atlas and axis to the large body and long neck of the giant rhino already characterized *P. grangeri* and *P. bugtiense*, and was further developed in *P. linxiaense*, whose atlas is elongated, indicative of a long neck and higher axis with a nearly horizontal position of its posterior articular face. There correlate to a more flexible neck[17,19–21]. *Paraceratherium lepidum* has an atlas resembling that of *P. bugtiense*, and an axis resembling that of *P. linxiaense*. In terms of limb bones, *P. lepidum* has the most massive third metacarpal within the genus.

The giant rhino of western Pakistan is from the Oligocene pre-Siwalik Chitarwata Formation. Specimens represent a single species, *Paraceratherium bugtiense*, while the genus distributed across the Mongolian Plateau, northwestern China, and Kazakhstan north to the Tibetan Plateau is highly diversified. Our phylogenetic analysis indicates that all six species of *Paraceratherium* are sister to *Aralotherium* and form a clade in which

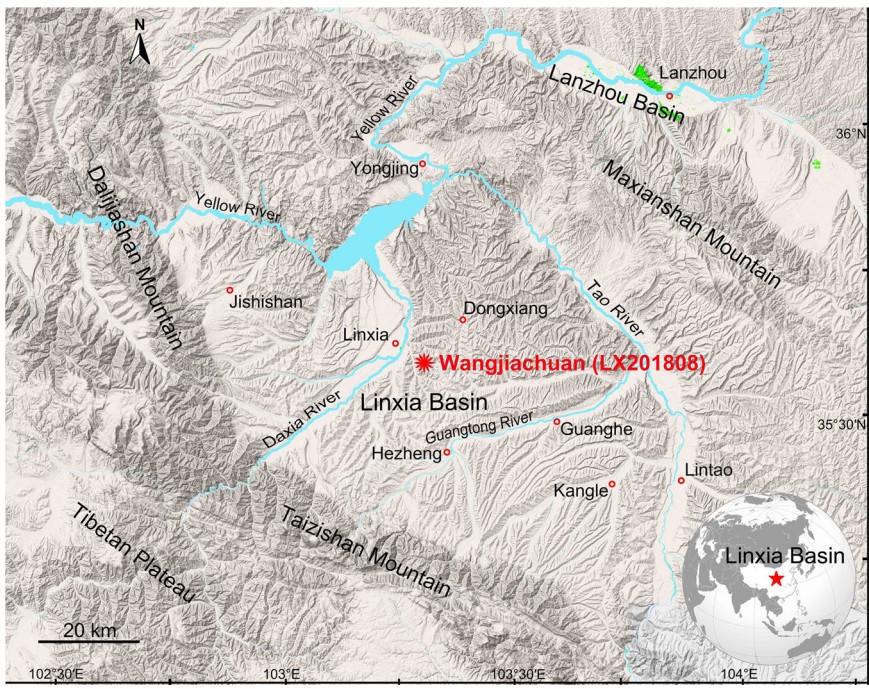

**Fig. 3 Type locality of *Paraceratherium linxiaense*.** Map showing the giant rhino fossil locality of the Linxia Basin in Wangjiachuan Village, Dongxiang County, Gansu Province, China.

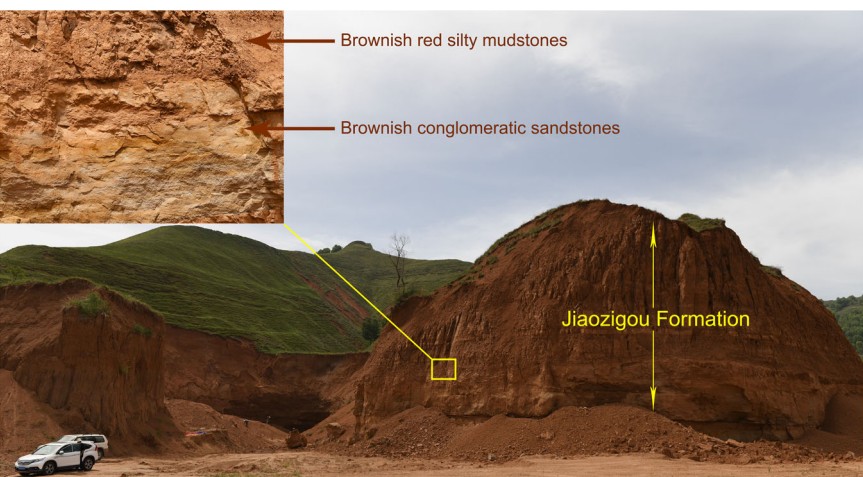

**Fig. 4 Type horizon of *Paraceratherium linxiaense*.** Exposures of fluvial and lacustrine sediments of the Linxia Basin (LX 1808) at the type locality of this new species (HMV 2006, yellow square) and where the axis and thoracic vertebrae were also found in Wangjiachuan Village, Dongxiang County, Gansu Province, China.

*P. grangeri* is the most primitive, succeeded by *P. huangheense* and *P. asiaticum*. In the early Oligocene, *P. asiaticum* dispersed westward to Kazakhstan from the hypothesized ancestral area of the genus *Paraceratherium* in Mongolia where *P. grangeri* originated, and its descendant lineage might have expanded to South Asia as *P. bugtiense*. In the late Oligocene, as the sister group of *P. bugtiense*, *P. lepidum* was found in Ningxia, Xinjiang and Kazakhstan, and *P. linxiaense* in Linxia. Our phylogenetic analysis places *P. bugtiense* of South Asia as a derivative of Central Asian *P. huangheense* via *P. asiaticum*, indicating dispersal southward through the Tibetan region. We note early Oligocene aridity in Central Asia at a time when South Asia was relatively moist, with a mosaic of forested and open landscapes[22,23]. Late Oligocene tropical conditions allowed the giant rhino to return northward to Central Asia, implying that the Tibetan region was

still not established as a high-elevation plateau. The largest land mammal might have been able to migrate freely, passing along eastern coast of the Tethys in the western part of Tibet, and even through some lowlands within what is now the plateau.

The paracerathere evolutionary trend can be analyzed against geological time. *P. grangeri* appeared in the Mongolian Plateau during the late early Oligocene with an age of about 31–28 Ma at Hsanda Gol, Mongolia[24], and it produced westward derivatives *P. huangheense* and *P. asiaticum*, reaching the Lanzhou Basin and the Turgai region in Kazakhstan, respectively[19,25]. *P. bugtiense* comes from the Oligocene Bugti Member of the lower Chitarwata Formation in South Asia[9,10,26]. Our cladistic analysis places the Oligocene *P. bugtiense* as sister taxon to the *P. lepidum* and *P. linxiaense* dichotomy. Late Oligocene northward dispersal of these two new forms is recorded as widespread fossils of

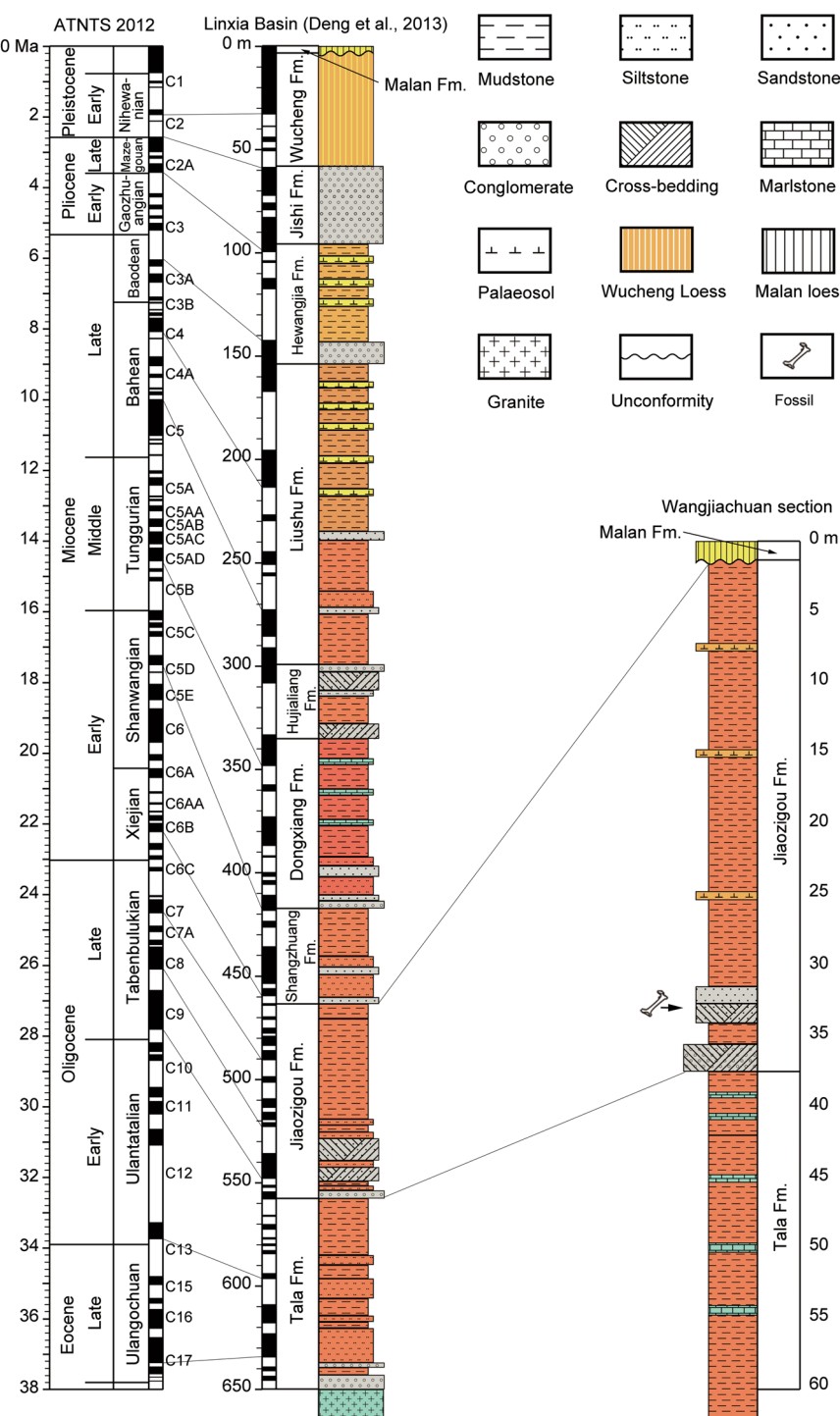

**Fig. 5 Cenozoic section of the Linxia Basin.** Deposits with correlation to ATNTS 2012[49] (left) for the magnetostratigraphic results[50] (middle) and the Wangjiachuan section bearing the fossils of *Paraceratherium linxiaense* sp. nov. (right).

*P. lepidum* distributed in the vast region between the Mongolian Plateau and Kazakhstan[7], and by *P. linxiaense* currently known only in the Linxia Basin.

Since the giant rhinos of Kazakhstan and Pakistan were contemporaries in the Oligocene, could they disperse directly between the two regions? Paleogeographically, it was impossible. In the Eocene, Asia was separated from Europe because the Middle East and Western Siberia were still occupied by the sea[27]. The Turgai Strait was still submerged but transitional terrestrial deposits were already developing in the Oligocene[25].

The southwest depression of the Tarim Basin extended eastward to reach the Altyn Tagh Mountain where the late Eocene to the early Miocene marine deposits are distributed and bear abundant marine foraminifers, ostracods, and bivalves[28–30]. Marine fish fossils, among which cartilaginous fishes strongly resemble the fish fauna of the Paris Basin[31], indicate an east-west sea in this area (Supplementary Note 3 and Supplementary Fig. 3). South Asia was separated from Kazakhstan by the sea, blocking giant rhinos from direct exchange in this direction (Fig. 7).

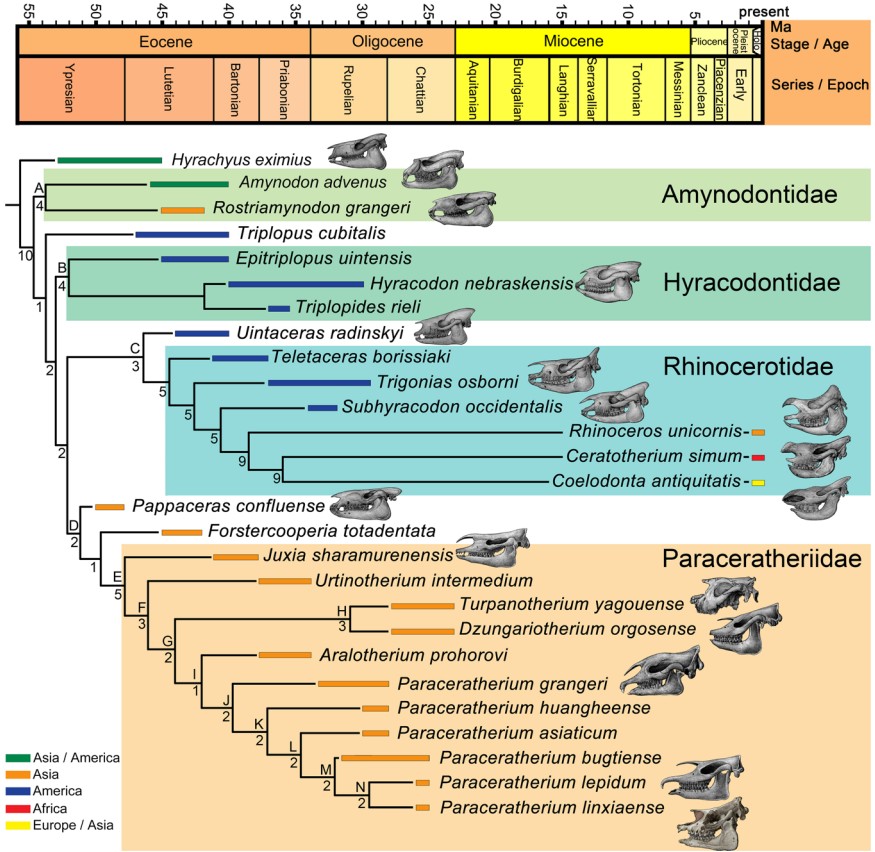

**Fig. 6 Phylogenetic relationship of giant rhinos.** Correlated with the geographical and geochronological distribution, based on the single most-parsimonious tree (length 327, consistency index 0.60, and retention index 0.79, with Bremer support values nearby the node).

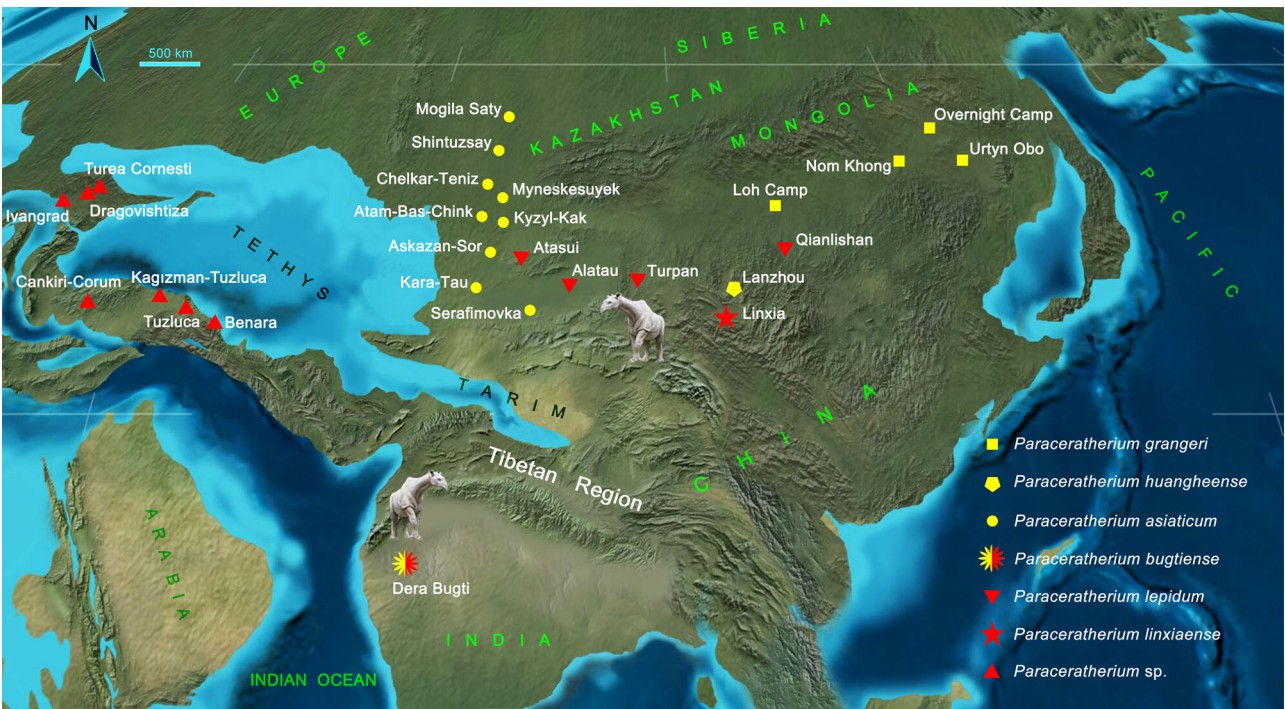

**Fig. 7 Distribution and dispersal of *Paraceratherium*.** Localities of the early Oligocene species were marked by the yellow color, and the red indicates the late Oligocene species. Dispersals of *Paraceratherium* between South Asia and other localities have to pass the Tibetan region, because most part of Central Asia, including southeastern Kazakhstan, Turpan Basin, and Tarim Basin was covered by the Tethys Ocean during the Oligocene. Paleogeography map is modified from Deep Time Maps (https://deeptimemaps.com) with license and a recent study[51].

During the Oligocene, dispersal for the giant rhino from the Mongolian Plateau to South Asia could have been along the eastern coast of the Tethys like other mammals, such as anthracotheres[32] and ruminants[33] as early as in the late Eocene or around the Eocene-Oligocene transition, or through some low-altitude areas in the hinterland of Tibet, demonstrating that the Tibetan region was not yet an elevated plateau as it is today, without height great enough to obstruct the dispersal of large mammals like the giant rhino. The topographical possibility that the giant rhino may have passed through the Tibetan region to reach the Indian-Pakistani subcontinent in the Oligocene can be supported by other evidence. The fish and plant fossils discovered from the Oligocene lacustrine deposits in the central Tibetan region display tropical characteristics, indicating an elevation of lower than 2000 m for this region[34−36]. Through to the late Oligocene, the evolution and dispersal from *P. bugtiense* to *P. linxiaense* and *P. lepidum* demonstrate that Tibet, as a plateau, did not yet exist and was not yet a barrier to exchange of largest land mammals.

## Methods

**Nomenclature Acts.** This published work and the nomenclatural acts have been registered in ZooBank, the proposed online registration system for the International Code of Zoological Nomenclature (ICZN). The ZooBank LSIDs (Life Science Identifiers) can be resolved and the associated information viewed through any standard web browser by appending the LSID to the prefix "http://zoobank.org/". The LSID for this publication is: urn:lsid:zoobank.org:pub:04E1C419-F1C1-4876-ACE9-3DC199F03696.

**Statistics and Reproducibility.** Based on the many studies on giant rhinos, especially the detailed discussion and recommendations[7], we consider *Dzungariotherium* and *Aralotherium* as valid genera, but *Indricotherium* and *Baluchitherium* as junior synonyms of *Paraceratherium*. There were several revisions at species level[7,14,21]. *Benaratherium* is not taken into account in this study because its limited materials are not accessible. In this first phylogeny analysis of Paraceratheriidae, we selected all 6 species of *Paraceratherium* and the type species of all other genera of this family[7], with the exception of *Turpanotherium*, which is represented by *T. yagouense* skull material instead of the less well represented type species *T. elegans*. Other Paleogene rhinocerotoid sister taxa, including two genera of Amynodontidae, four genera of Hyracodontidae and four of Rhinocerotidae, were added to test their relationship with Paraceratheriidae. In particular, with respect to Rhinocerotidae the extant species, plus 3 genera from Neogene and Quaternary were used to further reveal trends. *Hyrachyus* was used as outgroup. Whenever possible, we selected taxa with well-preserved samples and distinct diagnoses and morphology, especially taxa previously used in phylogenetic analyses to make well-tested comparisons. The samples of each taxon used in this study to code the characters were listed in Supplementary Data 1.

Large body size is the most outstanding feature of Paraceratheriidae, especially in the derived genera. In this study, we described and defined 73 new characters based on careful investigations and comparisons of giant rhinos, particularly the postcranial bones of derived paraceratheriids, including 11 of cranium and mandible, 22 of teeth, and 40 of postcranial bones. Additionally, we used the previously built 82 characters to cover the morphology of other Paleogene rhinoceroses and derived rhinoceros list groups, but with new data set and descriptions[16,37–40]. All characters used in this study appear in Supplementary Data 2.

In total, 155 characters were scored for the 27 taxa (Supplementary Data 3), all equally weighted and non-additive. The phylogenetic analysis was performed using TNT 1.5 in traditional search with the strategy of WAG and TBR, 100 random seed[41] (Supplementary Fig. 1). In order to obtain all the most parsimonious trees, the analysis was performed with 1000 replications, each of which got the best score. A single most parsimonious tree was attained with length 327, CI 0.60, RI 0.79 (Fig. 6). Bremer support values were calculated also by TNT with the TBR in 10 steps.

## Relationships of the family Paraceratheriidae.

Amynodontids were once considered as a sister group to rhinocerotids or paraceratheriids in previous work[16,42,43]. In our analysis, the amynodontid clade (*Amynodon* and *Rostriamynodon*) (node A in Fig. 6) is basal, following the outgroup *Hyrachyus*, with high Bremer support values (4) and four synapomorphies (1(1), shallow nasal notch anterior to P2; 47(2) and 63(2), highly enlarged upper and lower canines; 88 (1), absence of the first lower premolar p1), the latter three at greatly derived level make it differ from all other Paleogene rhinocerotoids. This result corresponds to the suggested Amynodontidae phylogeny[37,44].

Phylogeny of Hyracodontidae has long been debated but remains hardly resolved. Many authors hypothesized a closer relationship of Hyracodontidae and Paraceratheriidae[7,15,37,45], and several assigned it as a stem group basal to other rhinocerotids[16,42,43,46]. In this analysis, however, *Triplopus* was presented as a stem group sister to a "hyracodontid clade" of three genera *Epitriplopus*, *Hyracodon*, and *Triplopides*, which is not supported by postcranial features but is by four dental features (node B in Fig. 6): 63(1), lower canine is reduced to incisor size; 68(2), hypocone and protocone on P2 isolated and not connected with each other; 71(1), hypocone and protocone on P4 isolated but connected; 76(1), posterior part of the ectoloph on M1–2 is straight. The failure of this study to find a monophyletic hyracodontid group is not too surprising, due to its unresolved relationship with *Uintaceras*, *Pappaceras* and *Forstercooperia* (the latter two taxa were united with the paraceratheriids), as well as the limited taxa and postcranial bones of hyracodontid here for this family. The relationship of Hyracodontidae is therefore tentative and should be reassessed in the future studies.

Contrary to the recent previous analysis[44], our analysis of Rhinocerotidae found a closer relationship with Paraceratheriidae, sharing four synapomorphies (14(1), postglenoid foramen absent; 32(1), mental foramen at level of p2–p4; 110(1), robust humerus; 128(1), proximal articular facet in McIII high and evident). The significance of the absence of postglenoid foramen in the phylogeny of both families has been highlighted[14]. The posteriorly retracted mental foramen in the synapomorphies must be correlated with the enlarged lower incisor and mandibular symphysis, although each has different incisor specialization. This relationship of families partly reflects the ample postcranial samples of the taxa. The enlarged incisors of Rhinocerotidae had not yet appeared in *Uintaceras*, which therefore was not referred to this family although often has been considered as the closest primitive sister group[39,47]. It shares four synapomorphies with Rhinocerotidae clade (node C in Fig. 6) (24(1), occipital condyle with a median ridge; 83(1), M1-M2 with antecrochet; 114(0), shorter radius and ulnar relative to humerus; 155(1), MtIV with higher position relative to MtIII). The latter two characters listed in this node illustrate a unique evolution pattern to robust legs, not yet observed in other rhinocerotoids, shedding new light on the phylogeny of Rhinocerotidae.

The similarities of cheek teeth and skulls between *Forstercooperia* and *Juxia* have been discussed several times[7,14,48]. Our result did not show a monophyly of Forstercooperiinae, but supported a close relationship of *Pappaceras* and *Forstercooperia*, and *Juxia* with derived giant rhinos (node D in Fig. 6), these genera share four synapomorphies (9(1), zygomatic arch with a lobe-shaped blade; 69(0), widest position of P3-P4 crown at middle part; 97(1) and 98(1), trigonid of lower molar with a nearly flat or round out wall, and nearly U-shaped outline in occlusal view). Among Paleogene rhinocerotoids, the lobe-shaped blade of zygomatic arch is only present on groups in this clade. The reversal of P3-P4 outline should affect the phylogeny of less specialized groups of rhinocerotoid, such as *Forstercooperia*, *Triplopus*, and *Uintaceras*, but discussion of this issue is beyond the scope of this paper.

The true giant rhino clade (node E in Fig. 6) was supported by thirteen synapomorphies (1(2), nasal notch retracted; 4(1), premaxilla and nasal contact absent; 12(1), infraorbital foramen below nasal notch; 32(1), mental foramen at level of p2–p4; 39(1), upper incisor I1 with large size; 44(1), upper incisors sagittally arranged; 50(1), upraised extension of lower incisor i1 absent; 51(1), downward extension of lower incisor i1; 52(1), lower incisor i1 enlarged; 63(1), lower canine reduced in size; 68(2), hypocone and protocone isolated and not connected with each other in P2; 69(2), widest position of P3-P4 at posterior part of crown; 70(1), hypocone and protocone isolated but connected with each other in P3), with strong Bremer support value of 5 steps. Among these synapomorphies, the shape of nose, premaxilla, and lower and upper first incisors, make the giant rhino differ from all other Paleogene rhinocerotoids as well as Forstercooperiinae (Char. 1, 4, 12, 50, 51, 52), and the posteriorly situated widest position on P3-P4 (Char. 69) has never been observed on either Paleogene or later rhinocerotoids. On node F (Fig. 6), our analysis put a feature that represents a typical evolutionary tendency of Paraceratheriidae, namely the highly specialized chisel-liked lower incisor i1 (54(2), 61(1)), which united *Urtinotherium* and derived giant rhinos, together with another synapomorphies (28 (1)).

Other than basal *Juxia* and *Urtinotherium*, the similarities shared by all derived Paraceratheriidae are the fully reduced of anterior teeth even the first lower premolars, these features are listed in node G (Fig. 6) (58(1), i2 absent; 60(1), i3 absent; 62(1), canine absent; 88(1), p1 absent). Following these changes are two clades with more diverse morphology. One consists of *Dzungariotherium* and *Turpanotherium*, and is supported by six synapomorphies (node H in Fig. 6) (13 (2), skull dorsal profile concave; 76(1), posterior part of ectoloph of M1–2 straight; 78(1), lingual cingulum of M1-M2 absent; 80(1), protocone constriction of M1-M2 remarkable; 81(1), hypocone constriction of M1-M2 present; 83(1), antecrochet of M1-M2 present). Another clade (node I) (*Paraceratherium* and *Aralotherium*) is united by two synapomorphies (53(2), highly enlarged i1; 57(0), enlarged i1 on both sides distant). Based on known materials, these differences between the four genera were enumerated[7], with which our results are partially consistent in terms of tendency of specialization: complicated occlusal pattern of cheek teeth of node H, and presence of enlarged incisors of node I (Fig. 6). There are other features not listed in the node but attributed to shape the different evolution patterns, which separated these two clades, such as the reduced first upper or lower incisors in *Dzungariotherium* and *Turpanotherium*, and absence of p2 in the latter. The

alternative interpretation for the morphological differences among these genera is that the loss of incisors and p1-p2 derived from the further reduction of the rostral region, that is to say, *Paraceratherium* is probably a sister group of the ancestor of *Aralotherium*, *Dzungariotherium*, and *Turpanotherium*. *Dzungariotherium* has the largest body size, most reduced rostral teeth, shortened diastema and symphysis. In *Aralotherium*, the most outstanding feature is the downward extension of lower incisor i1, and its body size is smaller than *Paraceratherium* and *Dzungariotherium*. But probably the smallest genus is *Turpanotherium* whose rostral area is also reduced but with less reduced i1. It should be noted that this pair of sister clades were united based on another hypothesis. The reduced i1 and smaller body size is not a reversal, but probably derived from a less specialized or enlarged lineage, particularly for a family whose distinct tendency is gigantism.

*Paceratherium* clade is supported by eleven synapomorphies (node J) (2(1), nasal notch retracted to P2-P3; 6(1), length of diastema longer than upper premolar; 13 (1), arched skull dorsal profile; 29(1), robust crest along diastema of symphysis; 55 (1), very strong cingulum in i1; 95(1), entoconid connected with hypoconid in p4; 130(1), posterior McII facet in McIII present; 131(1), posterior McIV facet in McIII absent; 139(1), astragalus trochlea extended downward; 145(1), outline of second calcaneus facet in astragalus rounded; 155(1), higher position of MtIV relative to MtIII. This result highlighted many unique features of *Paraceratherium* relative to other derived giant rhinos, such as rostral area (Char. 6, 29), proximal articular facet in McIII (Char. 130, 131), outline of astragalus (Char. 139, 145), but further supported reversals in several species and their phylogenetic relationship within *Paraceratherium* suggested in the above section of this study. *P. grangeri* appeared as the ancestral morphotype succeeded by the second node (K) including *P. huangheense* whose lower cheek teeth were less specialized and four other species, with two synapomorphies (83(1), antecrochet in M1-M2 present; 90(1), single root in lower premolar p2). The remaining species were placed in the third node (L) with three synapomorphies (67(1), much smaller size of P2 relative to P3-P4; 76(1), straight posterior part of ectoloph in M1-M2; 80(1), remarkable protocone constriction in M1-M2). *P. asiaticum* is more derived than *P. grangeri* in having a gradually isolated hypocone in P3-P4, but less derived than *P. bugtiense* because of the presence of upper premolar P1. *P. bugtiense*, *P. lepidum* and *P. linxiaense* comprised node M, with two synapomorphies (64(1), upper premolar P1 absent; 82 (1), remarkable hypocone constriction in M1-M2). The node N is supported by two synapomorphies (34(1), coronoid process of ramus little developed; 85(1), crochet in M1-M2 present). These last two species represent the most specialized lineage with large body size and complicated occlusal pattern of the upper cheek teeth.

Based on the same matrix (27 taxa and 155 characters in Supplementary Data 3), we also run the likelihood estimation of relationship of the giant rhino, using Mrbayes 3.2.7 (lset nst = 6, rates = invgamma, ngen = 10000000). Within clade of *Paraceratherium*, six species display a relationship similar to the result of parsimonious analysis, but the relationships among four late Oligocene genera of the giant rhino are not resolved. Relationship of the Paleogene rhinocerotoids is also ambiguous in the likelihood estimation (Supplementary Fig. 2).

**Reporting summary**. Further information on research design is available in the Nature Research Reporting Summary linked to this article.

## Data availability

All specimens (HMV 2006 and HMV 2007) are deposited at the Hezheng Paleozoological Museum in Hezheng County, Gansu Province, China. Supporting data (character list and data matrix) for phylogenetic analyses in this study are provided in Supplementary Information.

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

## Acknowledgements

We thank Z.-X. Qiu, B.-Y. Wang, Z.-D. Qiu, X.-M. Wang, and S.-K. Hou for discussion, X.-M. Fang for help in the field, Y. Chen for drawing, W. Gao for photographing, and W. Zhang for taking surface scanning. We thank R. Blakey for providing the Oligocene paleogeographical map, which was modified in Fig. 7 and Supplementary Fig. 3. Prof. Robert Spicer, Prof. Pierre-Olivier Antoine, and an anonymous reviewer are kindly acknowledged for their comments on the manuscript. This research was supported by the Chinese Academy of Sciences (XDB26000000, XDA20070203, QYZDY-SSW-DQC022, GJHZ1885), the National Natural Science Foundation of China (41430102), and the Second Comprehensive Scientific Expedition on the Tibetan Plateau (2019QZKK0705).

## Author contributions

T.D. designed the research project; T.D., S.W., W.H., and S.C. conducted the fieldwork; T.D., X.L., and D.S. performed the phylogenetic analyses; T.D., X.L., and L.J.F. reconstructed paleogeography; and T.D., X.L. and L.J.F. wrote the manuscript.

## Competing interests

The authors declare no competing interests.
