## [Peer Review File · Communications Biology]

Reviewers' comments:

Reviewer #1 (Remarks to the Author):

This is a description of a new species of giant rhinoceros, augmented by a cladistic analysis of the family here called the Paraceratheriidae (=Indricotheriidae Borissiak) and featuring an argument about the elevation of the Tibetan Plateau.

Under the present pandemic I'm unable to access my library and in any case poorly qualified to assess the cladistic analysis, which is the main contribution of this manuscript. Still, I must note a few things that bear on it. For example, *Baluchitherium grangeri* Osborn, 1923 is assigned to *Paraceratherium*, which may be reasonable, but the fact that it falls into species-level synonymy with *Indricotherium transouralicum* Pavlova, 1922 is ignored. Overall, the relevant Russian literature is largely ignored. I suspect that incorporating it would improve the analysis and Gromova 1959 (cited) would be a good starting point.

The idea that indricotheres could provide evidence for the low elevation of the Tibetan Plateau in the Oligocene is diluted by the fact that stronger evidence already exists, some of which is even cited. And in any case I do not understand why the indricotheres could not have dispersed along the low-altitude eastern coast of the Tethys, rather than across the plateau. If the authors have an argument against the coastal route they do not mention it.

I was sad to see Fortelius & Kappelman (1993) cited for the overly high body mass estimates of indricotheres that the paper was intended to demolish. According to them, 11 tonnes is a good estimate of mean size while 15 tonnes might be a maximum value (an earlier paper with a similar conclusion often ignored is Gingerich (1990)). Fortelius & Kappelman also explain the trivial reason for the exaggerated estimate of Alexander (1989). This is very much a side issue in the manuscript but it is unfortunate that the abstract opens with a flagrant mis-citation.

Reviewer #2 (Remarks to the Author):

The paper entitled "A new Oligocene giant rhino and the migration of its lineage across Tibet" is an important contribution to our understanding of the extraordinary fauna that occupied the Tibetan area during the Paleogene. Our understanding of the contribution of the Paleogene Tibetan biota to the evolution of Asian biodiversity is only just becoming apparent with stunning new fossil finds across the modern plateau and the surrounding areas. The description is comprehensive and well done, but this group have an excellent reputation in this field, so that is not unexpected.

Where the paper is weak is in the assertion that the rhino's crossed the 'plateau' instead of circumventing the obstacles that the complex topography of the region presented at that time (Oligocene). It has been clear for some time that Tibet is an amalgam of terranes that accreted to Eurasia in the Mesozoic (see references in Kapp and DeCelles, 2019; Guillot et al., 2019) and that this produced a complex landscape even before the arrival of India. High E-W trending mountain ranges existed from early in the Cenozoic and by mid Eocene times the Gangdese and Qiantang highlands exceeded 4.5 km (Ding et al., 2014, 2017; Xu et al., 2013; Xu et al., 2018). Also in mid Eocene times, subtropical floras existed between these highlands (Tang et al., 2019; Liu et al., 2019; Del Rio et al., 2019) suggesting a lowland area and this great central valley persisted to the Oligocene (Wu et al., 2017; Su et al., 2019a). By the end of the Eocene the eastern end of the valley had achieved near modern elevations (~ 3.8 km, Su et al., 2019b). This created a complex, recently uplifted and therefore rugged, topography in the Oligocene that would be difficult to cross with suitable food resources for large mammals travelling in a N-S direction throughout the Paleogene, or even E-W by the late Oligocene. Please do not refer to it as a 'plateau'. Referent instead to the 'area now occupied by the plateau' or some such phrase. It clearly is not a plateau in the imaginative reconstruction in Fig. 3.

What I would like to see in this paper is the discussion of Tibetan migration routes that incorporates these new data, some of which are already acknowledged in the existing manuscript, but presented in such a way that does not suggest a low plateau across Tibet in the Oligocene. Unfortunately, such a flat landscape did not exist despite conceptual models utilising them (e.g. Botsyun et al. 2019). The most recent such effort Botsyun et al., 2019) has been shown to be methodologically flawed and was 'tested' using inappropriately aggregated and poorly constrained proxy data. See Valdes et al., 2019 for more detail. The Botsyun paper is not 'evidence of past reality, but a 'conceptual experiment'.

What lowlands that did exist in the Tibetan region were surrounded by high migration barriers. Strangely this is shown in Figure 3 where the migration arrow crosses the nearly 5 km high Gangdese. No justifications are given for this landscape reconstruction, which may have some elements of past 'truth' but we do not know what the evidence is underpinning the reconstruction. Where are the data to support it? They should be referenced. The area currently occupied by the Tibetan Plateau is labelled in Fig 3 as a plateau but clearly it is not. This term should be qualified. New data are emerging that the Oligocene in northern Tibet hosted at least some suitably vegetated landscapes (Song et al., 2020) through which migration could have taken place, while the Himalaya to the south were not yet high, but building against the Gangdese highland, so there would have been a lowland southern route also, albeit a more circuitous one. Please consider these alternative scenarios. The topography shown in Fig. 3 must be largely imagination because the data simply do not exist to provide such spatial detail. Again, this needs to be made clear somewhere.

This paper obviously presents a spectacular new fossil find, albeit not on the Tibetan Plateau proper, and deserves to be published in a high-profile journal. However, it could be improved with a discussion of the implications that is more grounded in recent literature and with more rigorous considerations of the past landscape.

The supplementary data are very long and, while generally well written, I feel could be shortened without loss of information. The dating is slightly unsatisfactory it that there appear to be no radiometric tie points in the magnetostratigraphy, and as we have seen throughout the Tibetan region the accepted dates, even when based partly on magnetostratigraphy, can often be in error by ~ 20 million years (Su et al., 2019b; Linnemann et al. 2018; Gourbet et al., 2017). This makes me inherently suspicious of dating without radiometric tie-points.

References:

Botsyun, S., et al., (2019) *Science* 363, eaaq1436 (2019). doi:10.1126/science.aaq1436
Del Rio, C., et al. (2020) *Amer. J. Bot.*, 107(1), 1–13, doi:10.1002/ajb2.1418.

Ding, L., et al., (2017) *Geology*, 45, 215–218.

Ding, L., et al. (2014) *Earth and Planetary Science Letters* 392, 250–264.

Gourbet, L., et al., 2017, *Tectonophysics* 700: 162–179,

<https://doi.org/10.1016/j.tecto.2017.02.007>.

Guillot, S., et al. (2019) *Russian Geology and Geophysics*, 60, 957–977.

Kapp, P., & DeCelles, P. G. (2019) *American Journal of Science*, 319, 159–254, doi:10.2475/03.2019.01.

Linnemann, U. et al. (2018). *Geology* 46: 3–6. <https://doi.org/10.1130/G39693.1>

Liu, J., et al. (2019). *Palaeogeogr., Palaeoclimatol., Palaeoecol.*, 524, 33–40.

Song, B. W. et al. (2020). *Earth and Planet. Sci. Lettrs.*, 537

<https://doi.org/10.1016/j.epsl.2020.116175>

Su, T., et al. (2019a) *Science Advances*, 5, eaav2189.

Su, T., et al. (2019b) *National Science Review*, doi: 10.1093/nsr/nwy062.

Tang, H., et al. (2019) *Journal of Systematics and Evolution*, 57(6), 670–677, doi:<https://doi.org/10.1111/jse.12505>.

Valdes, P.J., et al. (2019) *Science* 10.1126/science.aax8474.

Wu, F., et al. (2017). *Scientific Reports*, 7, 878-885, doi:10.1038/s41598-017-00928-9

Xu, Q., et al. (2013) *Earth and Planet. Sci. Lett.*, 362, 31-42.

Xu, Q., et al. (2018) *Gondwana Research*, 54, 50-61.

Bob Spicer

Reviewer #3 (Remarks to the Author):

This manuscript is devoted to the description of new remains of a giant rhino (Paraceratheriidae) from the Oligocene of Gansu Province, China. Phylogenetic analyses are performed (using parsimony and Bayesian methods). They further allow for a palaeogeographical scenario to be proposed.

The material is splendid and the description is thorough. Phylogenetic rationale is fully OK and relationships are well established with the current taxonomic sample.

Nevertheless, I am not overconfident in the dispersal scenario across the Tibetan Plateau, as the inclusion of western giant rhino material (from Balkans, Georgia, and Turkey) is likely to considerably change palaeogeographical inferences.

A number of issues need to be addressed specifically. They are appearing as notes in the annotated pdf. The most important ones are listed by order of appearance here below:

Title: I would recommend replacing "migration" by "dispersal" (more neutral and not season-related);

Abstract: There are a bunch of shortcuts, such as the lack of a systematic assignment, restricted ecology (it was not only living under arid conditions), overestimated claims on body-weight.

Line 20: Eastern European remains are not "questionable" and there are much more specimens and localities than mentioned in the reference cited (see Antoine et al., 2008 [*Zool J Linn Soc*] and Sen et al., 2011 [*NaturWiss*] for a review).

Line 24: Fossil record is quite biased in the area (there are undescribed remains in Iran and in Afghanistan, and the manuscript does not mention well-documented occurrences from Georgia and Turkey either, which has broad implications on dispersal scenarios).

Line 30 : In what the new species is more advanced than its sister species, *P. lepidum*? Moreover, *Paraceratherium bugtiense* has close relationships with this clade, not only with *P. linxiaense* nov. sp.

Line 34: What do you mean: *P. bugtiense* was the direct ancestor of other species? This is not plausible, as it has autapomorphies and as such it cannot be equated to their common ancestor.

Diagnosis: the small size of i1 may be a peculiar feature of *P. linxiaense* or it may document sexual dimorphism (as recognised on incisors in *P. bugtiense*; see Antoine et al., 2004 [op. cit.]).

Similarly, body size cannot be considered a clue, especially based on such a small sample (two individuals). Intraspecific (and intrapopulational) variability is extreme in giant rhinos.

Lines 61-133: where does the diagnosis end and the comparative description begin?

Lines 176-204: *Paraceratherium bugtiense* has a much longer range in the Bugti Hills, Pakistan than shown in the MPT. It was documented in all fossil-yielding levels of the Bugti Member of the Chitarwata Formation (see Welcomme et al., 2001 [op. cit.]; Antoine et al., 2003 [*Can J Earth Sci*]; Métais et al., 2009 table 2 [*J Asian Earth Sci*], i.e. spanning the entire Oligocene epoch. This has implications on inferred ages for several deeper nodes and for dispersal scenarios, especially when taking into account occurrences to the west (Afghanistan, Iran, Georgia, Turkey, and Balkans). Aside from Balkans and eastern European records, *Paraceratherium* was documented in Caucasus (Benara, Georgia c. 27 Ma; under the name *Benaratherium*; Gabunia, various works; Métais et al., 2016 for the age [*Palaeont Electr*]) and Anatolia (Turkey: Gözüzilli and other localities of the Kizilirmak Fm. in Central Anatolia (28-24 Ma): Antoine et al., 2008 [*Zool J Linn Soc*]; Métais et al., 2016 [*Palaeont Electr*]; Tuzluca, close to the Armenian border (late Oligocene):

Sen et al., 2011 [NaturWiss])

Lines 205-228: There is no argument for discarding a complete Peri-Tethyan pathway for giant rhinos between Balochistan, Iran, Caucasus, Anatolia, Balkans, Kazakhstan, and China around the Eocene-Oligocene Transition and in the early Oligocene. Such a palaeobiogeographical hypothesis is also supported by rhinocerotids and anthracotheriids (Saraç, 2001 [DEINSEA]; Böhme et al., 2014 [Zitteliana]), by ruminants (Métais et al., 2009 [J Asian Earth Sci], 2016 [Palaeont Electron]; Mennecart et al., 2019 [Palaeo3]) and other ungulates (e.g., entelodontids or suids). Most of these references provide Oligocene paleogeographic maps, likely to provide alternate scenarios for giant rhino (and associated ungulate) dispersals. The map as provided in Fig. 3 is only a snapshot in a much geographically-unstable area due to both tectonics and eustasy, and dispersals were probably diachronous, following local environmental changes.

It has been a pleasure to read his manuscript, and I am eager to reading a revised version (and then the published article)!

Pierre-Olivier Antoine

Revision Notes

Response to Reviewer #1's comments:

Comments: This is a description of a new species of giant rhinoceros, augmented by a cladistic analysis of the family here called the Paraceratheriidae (=Indricotheriidae Borissiak) and featuring an argument about the elevation of the Tibetan Plateau.

Under the present pandemic I'm unable to access my library and in any case poorly qualified to assess the cladistic analysis, which is the main contribution of this manuscript. Still, I must note a few things that bear on it. For example, *Baluchitherium grangeri* Osborn, 1923 is assigned to *Paraceratherium*, which may be reasonable, but the fact that it falls into species-level synonymy with *Indricotherium transouralicum* Pavlova, 1922 is ignored. Overall, the relevant Russian literature is largely ignored. I suspect that incorporating it would improve the analysis and Gromova 1959 (cited) would be a good starting point.

Response: Gromova (1959) considered that Borissiak published *Indricotherium asiaticum* in 1923 and Pavlow established *I. transouralicum* in 1922, so she believed that the latter was prior. On the other hand, *Indricotherium asiaticum* n. g. n. sp. appeared actually in 1918 in a monograph about *Epiaceratherium turgaicum* (Borissiak, 1918, p. 69), and Borissiak also indicated that the description and definition of this species came from an earlier publication [Comtes Rendu, V. 162, No. 14, 3 avr. 1916, Mem. Ac. Sc. Petrogr. XXXVI]. According to International Code of Zoological Nomenclature 4th Edition, as a result, *I. asiaticum* is a valid species and *I. transouralicum* is its junior synonymy.

Borissiak A A, 1918. Osteology of *Epiaceratherium turgaicum* n. sp. Rus Paleont Soc Monogr, I: 1-84.

Borissiak A A, 1923. *Indricotherium asiaticum* n. g., n. sp. Mem Soc Geol France Paleont, XXV, fasc. 3(59): 1-15.

Pavlow M, 1928. *Indricotherium transouralicum*, n. sp. provenant du district de Tourgay. Bull Soc Natur, Moscow, n s, 31: 95-116.

Comments: The idea that indricotheres could provide evidence for the low elevation of the Tibetan Plateau in the Oligocene is diluted by the fact that stronger evidence already exists, some of which is even cited. And in any case I do not understand why the indricotheres could not have dispersed along the low-altitude eastern coast of the Tethys, rather than across the plateau. If the authors have an argument against the coastal route they do not mention it.

Response: We have revised the main dispersal route of giant rhinos along the eastern coast of the Tethys.

Comments: I was sad to see Fortelius & Kappelman (1993) cited for the overly high body mass estimates of indricotheres that the paper was intended to demolish. According to them, 11 tonnes is a good estimate of mean size while 15 tonnes might be a maximum value (an earlier paper with a similar conclusion often ignored is Gingerich (1990)). Fortelius & Kappelman also explain the trivial reason for the exaggerated estimate of Alexander (1989). This is very much a side issue in the manuscript but it is unfortunate that the abstract opens with a flagrant mis-citation.

Response: We have cancelled the outrageous overestimates of giant rhinos.

Response to Reviewer #2's (Prof. Robert Spicer) comments:

Comments: The paper entitled "A new Oligocene giant rhino and the migration of its lineage across Tibet" is an important contribution to our understanding of the extraordinary fauna that occupied the Tibetan area during the Paleogene. Our understanding of the contribution of the Paleogene Tibetan biota to the evolution of Asian biodiversity is only just becoming apparent with stunning new fossil finds across the modern plateau and the surrounding areas. The description is

comprehensive and well done, but this group have an excellent reputation in this field, so that is not unexpected.

Where the paper is weak is in the assertion that the rhino's crossed the 'plateau' instead of circumventing the obstacles that the complex topography of the region presented at that time (Oligocene). It has been clear for some time that Tibet is an amalgam of terranes that accreted to Eurasia in the Mesozoic (see references in Kapp and DeCelles, 2019; Guillot et al., 2019) and that this produced a complex landscape even before the arrival of India. High E-W trending mountain ranges existed from early in the Cenozoic and by mid Eocene times the Gangdese and Qiantang highlands exceeded 4.5 km (Ding et al., 2014, 2017; Xu et al., 2013; Xu et al., 2018). Also in mid Eocene times, subtropical floras existed between these highlands (Tang et al., 2019; Liu et al., 2019; Del Rio et al., 2019) suggesting a lowland area and this great central valley persisted to the Oligocene (Wu et al., 2017; Su et al., 2019a). By the end of the Eocene the eastern end of the valley had achieved near modern elevations (~ 3.8 km, Su et al., 2019b). This created a complex, recently uplifted and therefore rugged, topography in the Oligocene that would be difficult to cross with suitable food resources for large mammals travelling in a N-S direction throughout the Paleogene, or even E-W by the late Oligocene. Please do not refer to it as a 'plateau'. Referent instead to the 'area now occupied by the plateau' or some such phrase. It clearly is not a plateau in the imaginative reconstruction in Fig. 3.

Response: We have revised the text and indicated that the giant rhino circumvented the area now occupied by the Tibetan plateau along the eastern coast of the Tethys. We did not refer to the Paleogene Tibet as a plateau.

Comments: What I would like to see in this paper is the discussion of Tibetan migration routes that incorporates these new data, some of which are already acknowledged in the existing manuscript, but presented in such a way that does not suggest a low plateau across Tibet in the Oligocene. Unfortunately, such a flat landscape did not exist despite conceptual models utilising them (e.g. Botsyun et al.

2019). The most recent such effort (Botsyun et al., 2019) has been shown to be methodologically flawed and was 'tested' using inappropriately aggregated and poorly constrained proxy data. See Valdes et al., 2019 for more detail. The Botsyun paper is not 'evidence of past reality, but a 'conceptual experiment'.

Response: We have revised the text and indicated that the eastern coast of the Tethys at the western margin of the modern Tibetan Plateau was the main dispersal route of the giant rhino. We also indicated that the conceptual experiment of Botsyun et al. (2019) was intensely debated.

Comments: What lowlands that did exist in the Tibetan region were surrounded by high migration barriers. Strangely this is shown in Figure 3 where the migration arrow crosses the nearly 5 km high Gangdese. No justifications are given for this landscape reconstruction, which may have some elements of past 'truth' but we do not know what the evidence is underpinning the reconstruction. Where are the data to support it? They should be referenced. The area currently occupied by the Tibetan Plateau is labelled in Fig 3 as a plateau but clearly it is not. This term should be qualified. New data are emerging that the Oligocene in northern Tibet hosted at least some suitably vegetated landscapes (Song et al., 2020) through which migration could have taken place, while the Himalaya to the south were not yet high, but building against the Gangdese highland, so there would have been a lowland southern route also, albeit a more circuitous one. Please consider these alternative scenarios. The topography shown in Fig. 3 must be largely imagination because the data simply do not exist to provide such spatial detail. Again, this needs to be made clear somewhere.

Response: We have cancelled the migration arrow across Tibet, and changed “Tibetan Plateau” into “Tibetan region” in Fig. 3. In the text, we have indicated that the giant rhino dispersed mainly along the eastern coast of the Tethys at the western part of Tibet or through some lowlands of this region. The landscape reconstruction during the Oligocene is from Deep Time Maps (<https://deeptimemaps.com>). We have asked the designer to revise the topography of Tibet based on Spicer et al. (2020) and other recent publications, showing the lower Himalayas and the higher Gangdese.

Comments: This paper obviously presents a spectacular new fossil find, albeit not on the Tibetan Plateau proper, and deserves to be published in a high-profile journal. However, it could be improved with a discussion of the implications that is more grounded in recent literature and with more rigorous considerations of the past landscape.

Response: We have added the recent research of Spicer et al. (2020) about the paleo-altitude reconstruction of the Tibetan Plateau.

Comments: The supplementary data are very long and, while generally well written, I feel could be shortened without loss of information. The dating is slightly unsatisfactory in that there appear to be no radiometric tie points in the magnetostratigraphy, and as we have seen throughout the Tibetan region the accepted dates, even when based partly on magnetostratigraphy, can often be in error by ~ 20 million years (Su et al., 2019b; Linnemann et al. 2018; Gourbet et al., 2017). This makes me inherently suspicious of dating without radiometric tie-points.

Response: We have shortened the supplementary data, especially the description about vertebrae and paleogeography. Although the magnetostratigraphy of the Linxia Basin has no radiometric tie points, it has many fast-evolved rodent fossils as time markers for constraints, so the paleomagnetic datings are accurate.

Response to Reviewer #3's (Prof. Pierre-Olivier Antoine) comments:

Comments: This manuscript is devoted to the description of new remains of a giant rhino (Paraceratheriidae) from the Oligocene of Gansu Province, China. Phylogenetic analyses are performed (using parsimony and Bayesian methods). They further allow for a palaeogeographical scenario to be proposed.

The material is splendid and the description is thorough. Phylogenetic rationale is fully OK and relationships are well established with the current taxonomic sample.

Nevertheless, I am not overconfident in the dispersal scenario across the Tibetan Plateau, as the inclusion of western giant rhino material (from Balkans, Georgia, and Turkey) is likely to considerably change palaeogeographical inferences.

Response: We have revised the dispersal scenario of the giant rhino.

Comments: A number of issues need to be addressed specifically. They are appearing as notes in the annotated pdf. The most important ones are listed by order of appearance here below:

Response: We have revised our manuscript according to the notes in the annotated pdf.

Comments: Title: I would recommend replacing “migration” by “dispersal” (more neutral and not season-related);

Response: We have replaced “migration” by “dispersal”.

Comments: Abstract: There are a bunch of shortcuts, such as the lack of a systematic assignment, restricted ecology (it was not only living under arid conditions), overestimated claims on body-weight.

Response: We have added the systematic assignment and restricted ecology, and cancelled the outrageous overestimates of the giant rhino.

Comments: Line 20: Eastern European remains are not “questionable” and there are much more specimens and localities than mentioned in the reference cited (see Antoine et al., 2008 [Zool J Linn Soc] and Sen et al., 2011 [NaturWiss] for a review).

Response: We have cancelled “questionable” and cited Antoine et al. (2008) and Sen et al. (2011) for Anatolia and Gabunia (1964) for Caucasus as references.

Comments: Line 24: Fossil record is quite biased in the area (there are undescribed remains in Iran and in Afghanistan, and the manuscript does not mention

well-documented occurrences from Georgia and Turkey either, which has broad implications on dispersal scenarios.

Response: We have added Anatolia and Caucasus as the giant rhino's distribution.

Comments: Line 30: In what the new species is more advanced than its sister species, *P. lepidum*? Moreover, *Paraceratherium bugtiense* has close relationships with this clade, not only with *P. linxiaense* nov. sp.

Response: We have revised this sentence as “its clade with *P. lepidum* has a tight relationship to *P. bugtiense*”.

Comments: Line 34: What do you mean: *P. bugtiense* was the direct ancestor of other species? This is not plausible, as it has autapomorphies and as such it cannot be equated to their common ancestor.

Response: We have revised this sentence as “as the sister group of *P. bugtiense*, *P. lepidum* was found in Ningxia, Xinjiang and Kazakhstan, and *P. linxiaense* in Linxia.”

Comments: Diagnosis: the small size of i1 may be a peculiar feature of *P. linxiaense* or it may document sexual dimorphism (as recognised on incisors in *P. bugtiense*; see Antoine et al., 2004 [op. cit.]. Similarly, body size cannot be considered a clue, especially based on such a small sample (two individuals). Intraspecific (and intrapopulation) variability is extreme in giant rhinos.

Response: We have added “small” in the description of i1, and cancelled the description of “with a basal cranial length of 1148 mm”.

Comments: Lines 61-133: where does the diagnosis end and the comparative description begin?

Response: We have added a subhead “Comparative description”.

Comments: Lines 176-204: *Paraceratherium bugtiense* has a much longer range in the Bugti Hills, Pakistan than shown in the MPT. It was documented in all fossil-yielding levels of the Bugti Member of the Chitarwata Formation (see Welcomme et al., 2001 [op. cit.]; Antoine et al., 2003 [Can J Earth Sci]; Métais et al., 2009 table 2 [J Asian Earth Sci], i.e. spanning the entire Oligocene epoch. This has implications on inferred ages for several deeper nodes and for dispersal scenarios, especially when taking into account occurrences to the west (Afghanistan, Iran, Georgia, Turkey, and Balkans). Aside from Balkans and eastern European records, *Paraceratherium* was documented in Caucasus (Benara, Georgia c. 27 Ma; under the name *Benaratherium*; Gabunia, various works; Métais et al., 2016 for the age [Palaeont Electr]) and Anatolia (Turkey: Gözüüzilli and other localities of the Kizilirmak Fm. in Central Anatolia (28-24 Ma): Antoine et al., 2008 [Zool J Linn Soc]; Métais et al., 2016 [Palaeont Electr]; Tuzluca, close to the Armenian border (late Oligocene): Sen et al., 2011 [NaturWiss])

Response: We have revised the stratigraphical distribution of *P. bugtiense* to cover the entire Oligocene epoch in the text and Fig. 2, and cited more publications. We added the geographical distribution of the genus *Paraceratherium* in Anatolia and Caucasus, and cited more publications.

Comments: Lines 205-228: There is no argument for discarding a complete Peri-Tethyan pathway for giant rhinos between Balochistan, Iran, Caucasus, Anatolia, Balkans, Kazakhstan, and China around the Eocene-Oligocene Transition and in the early Oligocene. Such a palaeobiogeographical hypothesis is also supported by rhinocerotids and anthracotheriids (Saraç, 2001 [DEINSEA]; Böhme et al., 2014 [Zitteliana]), by ruminants (Métais et al., 2009 [J Asian Earth Sci], 2016 [Palaeont Electron]; Mennecart et al., 2019 [Palaeo3]) and other ungulates (e.g., entelodontids or suids). Most of these references provide Oligocene paleogeographic maps, likely to provide alternate scenarios for giant rhino (and associated ungulate) dispersals. The map as provided in Fig. 3 is only a snapshot in a much geographically-unstable area

due to both tectonics and eustasy, and dispersals were probably diachronous, following local environmental changes.

Response: We have added the dispersal route of the giant rhino along the eastern coast of the Tethys like other mammals, such as anthracotheres (Böhme et al., 2013) and ruminants (Métais et al., 2017).

It has been a pleasure to read his manuscript, and I am eager to reading a revised version (and then the published article)!

Response: We have finished a substantially revised manuscript.

Reviewers' comments:

Reviewer #2 (Remarks to the Author):

This revised version is a significant improvement over the original and the authors have done a good job in addressing the reviewer comments. There are still issues with the writing, which is a little awkward in places, but I have suggested possible changes on the annotated pdf. Hopefully this will assist in preparing a version for publication.

The claims regarding movement through the Tibetan region are now more realistic in the light of current thinking about the palaeotopography, but these changes do make the inferences less dramatic (quite rightly). Tibet is still in the title, which could be interpreted by some as meaning a high plateau, so Tibetan region would be more honest, particularly as all the reviewers point out dispersal routes that circumvent the region that is now the plateau, but that is an editorial decision.

It would be wrong to assume the lowlands through the centre of what is now the plateau (basically along the Bangang-Nujiang Suture Zone) were below 2 km throughout all of the Oligocene as stated, because there is evidence of rapid regional surface rise beginning at ~ 26 Ma, and it may be that by the Miocene this part of central Tibet was above 3 km, but that has yet to be properly quantified. I have suggested qualifications to the text to get around this uncertainty.

With minor revisions this could be published as I no longer see major scientific flaws or omissions.

Bob Spicer

Reviewer #3 (Remarks to the Author):

I was lucky enough to act as a reviewer for the original submission. As stated in the rebuttal letter, the majority of reviewers' suggestions has been followed. Yet, it seems that some requests for major changes have not been taken into account satisfactorily in the revised version.

In particular neither the authors have edited the introduction, mostly copy-pasted from the abstract (and vice versa), nor they have reorganised the diagnosis. Adding a 'comparative description' subheading is not enough, as this section (lines 90-110) still goes back and forth from cranial to dental, then from lower to upper dental features.

I am not further convinced by the very long section devoted to dispersal routes and timing for *Paraceratherium* (lines 202-213): the concerned fossil record is far from being comprehensive and probably highly biased to the southeast, so an objective support is lacking. Perhaps adding hypothesised ancestral geographical areas could be of some help in this purpose.

I do not understand the sentence "*Paraceratherium* is probably an ancestor group of *Aralotherium*, *Dzungariotherium*, and *Turpanotherium*." (lines 376-377), as it goes against the topology of the most parsimonious tree.

Some references are fully distorted in their use: the reference 1 was already misused in the original version (two reviewers had highlighted this issue: the main result of Fortelius & Kappelman (1990) is the fact that *Paraceratherium* was NOT the largest land mammal ever – hence the title "the largest land mammal ever imagined"); the newly-added references 32 and 33 do conclude that the southwestern route i) might have been used by land mammals (anthracotheres and rhinoceroses) ii) as early as in the late Eocene or around the Eocene-Oligocene transition.

At last, I would once again recommend western occurrences of *Paraceratherium* to be added to the palaeogeographical map (Fig. 7: Asia Minor, Caucasus, Anatolia, and Balkans). Noteworthy, *Paraceratherium bugtiense* is mentioned as an early Oligocene species, whereas it ranges the entire Oligocene epoch.

Pierre-Olivier Antoine

Revision 2 Notes

Response to Reviewer #2's comments:

Comments: This revised version is a significant improvement over the original and the authors have done a good job in addressing the reviewer comments. There are still issues with the writing, which is a little awkward in places, but I have suggested possible changes on the annotated pdf. Hopefully this will assist in preparing a version for publication.

Response: We have revised our manuscript according to the annotated pdf. About the sentence on lines 204-205 that is an awkward sentence and is a repeat of that on lines 48-50, we deleted the sentence on lines 48-50.

Comments: The claims regarding movement through the Tibetan region are now more realistic in the light of current thinking about the palaeotopography, but these changes do make the inferences less dramatic (quite rightly). Tibet is still in the title, which could be interpreted by some as meaning a high plateau, so Tibetan region would be more honest, particularly as all the reviewers point out dispersal routes that circumvent the region that is now the plateau, but that is an editorial decision.

Response: We have revised “Tibet” in the title into “the Tibetan region”.

Comments: It would be wrong to assume the lowlands through the centre of what is now the plateau (basically along the Bangang-Nujiang Suture Zone) were below 2 km throughout all of the Oligocene as stated, because there is evidence of rapid regional surface rise beginning at ~ 26 Ma, and it may be that by the Miocene this part of central Tibet was above 3 km, but that has yet to be properly quantified. I have suggested qualifications to the text to get around this uncertainty.

Response: As qualifications, we have revised our statement and deleted the word “entire” from “entire Oligocene”: “The Tibetan region likely hosted some areas with low elevation, possibly under 2000 m during Oligocene, and the lineage of giant

rhinos could **have dispersed** freely along the eastern coast of the Tethys Ocean and **perhaps** through some lowlands of this region.” in the abstract. We also revised corresponding context in the main text.

Response to Reviewer #3’s comments:

Comments: I was lucky enough to act as a reviewer for the original submission. As stated in the rebuttal letter, the majority of reviewers’ suggestions has been followed. Yet, it seems that some requests for major changes have not been taken into account satisfactorily in the revised version. In particular neither the authors have edited the introduction, mostly copy-pasted from the abstract (and vice versa), nor they have reorganised the diagnosis. Adding a ‘comparative description’ subheading is not enough, as this section (lines 90-110) still goes back and forth from cranial to dental, then from lower to upper dental features.

Response: We have edited the introduction to make it different from the abstract, reorganised the diagnosis, and revised the comparative description.

Comments: I am not further convinced by the very long section devoted to dispersal routes and timing for *Paraceratherium* (lines 202-213): the concerned fossil record is far from being comprehensive and probably highly biased to the southeast, so an objective support is lacking. Perhaps adding hypothesised ancestral geographical areas could be of some help in this purpose.

Response: We have added the hypothesised ancestral geographical area of the genus *Paraceratherium* in Mongolia where *P. grangeri* originated.

Comments: I do not understand the sentence “*Paraceratherium* is probably an ancestor group of *Aralotherium*, *Dzungariotherium*, and *Turpanotherium*.” (lines 376-377), as it goes against the topology of the most parsimonious tree.

Response: We have revised this sentence.

Comments: Some references are fully distorted in their use: the reference 1 was already misused in the original version (two reviewers had highlighted this issue: the main result of Fortelius & Kappleman (1990) is the fact that *Paraceratherium* was NOT the largest land mammal ever – hence the title “the largest land mammal ever imagined”); the newly-added references 32 and 33 do conclude that the southwestern route i) might have been used by land mammals (anthracotheres and rhinoceroses) ii) as early as in the late Eocene or around the Eocene-Oligocene transition.

Response: We have revised our description to indicate the giant rhino has been considered as one of the largest land mammals. We have completed the quotation for references 32 and 33.

Comments: At last, I would once again recommend western occurrences of *Paraceratherium* to be added to the palaeogeographical map (Fig. 7: Asia Minor, Caucasus, Anatolia, and Balkans). Noteworthy, *Paraceratherium bugtiense* is mentioned as an early Oligocene species, whereas it ranges the entire Oligocene epoch.

Response: We have added the western occurrences of *Paraceratherium* and expanded the chorological range of *Paraceratherium bugtiense* in Fig. 7.

Reviewers' Comments:

Reviewer #2:

Remarks to the Author:

The authors have taken on board my earlier comments and from a scientific perspective I am happy with the manuscript. Our knowledge of the evolution of the Tibetan region has advanced greatly in the last couple of years and this is now reflected in the revised manuscript. Inevitably new issues with the use of English have crept in with the revision, and I have suggested changes on the annotated pdf. For clarity regarding these changes I have annotated the 'clean' version of the revised manuscript. Some of these changes may require editorial intervention. During the revision process other relevant papers documenting the existence of a Paleogene central Tibetan lowland, its vegetation and climate have been published. I am thinking here of Su et al. 2020 (PNAS. <https://doi.org/10.1073/pnas.2012647117>). If this paper is to be as current as possible when published such papers should be included. Again, this is an editorial decision.

Bob Spicer.

Response to the Reviewer's comments:

Comments: The authors have taken on board my earlier comments and from a scientific perspective I am happy with the manuscript. Our knowledge of the evolution of the Tibetan region has advanced greatly in the last couple of years and this is now reflected in the revised manuscript. Inevitably new issues with the use of English have crept in with the revision, and I have suggested changes on the annotated pdf. For clarity regarding these changes I have annotated the 'clean' version of the revised manuscript. Some of these changes may require editorial intervention. During the revision process other relevant papers documenting the existence of a Paleogene central Tibetan lowland, its vegetation and climate have been published. I am thinking here of Su et al. 2020 (PNAS. <https://doi.org/10.1073/pnas.2012647117>). If this paper is to be as current as possible when published such papers should be included. Again, this is an editorial decision.

Response: We have revised our manuscript according to the annotated pdf. Because the paper of Su et al. (2020, PNAS) is about the Eocene ecosystem in the Tibetan region, earlier than the age Oligocene of our manuscript, we do not cite it.